# Recovery priorities in degenerative cervical myelopathy: a cross-sectional survey of an international, online community of patients

Benjamin Davies,[1] Oliver Mowforth [ID],[1] Iwan Sadler,[2] Bizhan Aarabi,[3] Brian Kwon,[4] Shekar Kurpad,[5] James S Harrop,[6] Jefferson R Wilson,[7] Robert Grossman,[8] Michael G Fehlings,[7] Mark Kotter[1,9]

Global Spine Congress, Toronto, Canada, May 2019. World Federation of Neurosurgical Socieities, Belgrade, Serbia, March 2019. Joint meeting of The Society of British Neurological Surgeons and Association of British Neurologists, London, United Kingdom, September 2018.

For numbered affiliations see end of article.

**Correspondence to**
Dr Mark Kotter;
mrk25@cam.ac.uk

## ABSTRACT

**Objectives** To establish the recovery priorities of individuals suffering with degenerative cervical myelopathy (DCM).

**Design** A cross-sectional, observational study.

**Setting** Patients from across the world with a diagnosis of DCM accessed the survey over an 18-month period on Myelopathy.org, an international myelopathy charity.

**Participants** 481 individuals suffering from DCM completed the online survey fully.

**Main outcome measures** Functional recovery domains were established through qualitative interviews and a consensus process. Individuals were asked about their disease characteristics, including limb pain (Visual Analogue Scale) and functional disability (patient-derived version of the modified Japanese Orthopaedic Association score). Individuals ranked recovery domains (arm and hand function, walking, upper body/trunk function, sexual function, elimination of pain, sensation and bladder/bowel function) in order of priority. Priorities were analysed as the modal first priority and mean ranking. The influence of demographics on selection was analysed, with significance $p < 0.05$.

**Results** Of 659 survey responses obtained, 481 were complete. Overall, pain was the most popular recovery priority (39.9%) of respondents, followed by walking (20.2%), sensation (11.9%) and arm and hand function (11.5%). Sexual function (5.7%), bladder and bowel (3.7%) and trunk function (3.5%) were chosen less frequently. When considering the average ranking of symptoms, while pain remained the priority (2.6±2.0), this was closely followed by walking (2.9±1.7) and arm/hand function (3.0±1.4). Sensation ranked lower (4.3±2.1). With respect to disease characteristics, overall pain remained the recovery priority, with the exception of patients with greater walking impairment ($p < 0.005$) who prioritised walking, even among patients with lower pain scores.

**Conclusions** This is the first study investigating patient priorities in DCM. The patient priorities reported provide an important framework for future research and will help to ensure that it is aligned with patient needs.

## INTRODUCTION

Degenerative cervical myelopathy (DCM) has been coined as an umbrella term for

### Strengths and limitations of this study

► This is the largest study of patient perspective in degenerative cervical myelopathy (DCM) to date and the first to consider patient recovery priorities.

► This study is unique in reporting on both surgical and non-surgical DCM patients.

► This study includes a broad demographic representation of patients from across the globe and includes subgroup analysis.

► This is an open-access, internet-based survey, a methodology which can lead to a sampling bias.

► Efforts to mitigate against sampling bias, alongside reassuring subgroup analysis suggest this risk is low.

degenerative and congenital or acquired conditions of the cervical spine, such as spondylosis or ossification of the posterior longitudinal ligament, which lead to symptomatic cord compression.[1] With an estimated prevalence of up to 5% in individuals above 40 years old,[2 3] DCM is the most common cause of spinal cord dysfunction worldwide.[1] Given its degenerative aetiology and the rising age of the population, this incidence is expected to rise.[4]

The cervical spinal cord acts as a processor and conduit of information between the brain and the periphery. Its injury can, therefore, give rise to a range of possible symptoms.[1] These include pain, paraesthesia, weakness, unsteadiness, frequent falls, bladder or bowel dysfunction and impotence in men.[5] At early stages, individual symptoms may occur in isolation, but more typically occur in combination, especially as the disease advances.

At present, decompressive surgery is the only evidence-based treatment for DCM.[6] Surgical decompression is able to halt the progression of symptoms and offer limited,

although clinically-relevant[7] improvements across a range of domains.[8 9] However, due to the limited intrinsic capacity for the spinal cord to repair, most patients do not make a full recovery, and instead suffer lifelong disabilities.[9] As a consequence, unemployment and/or dependency is prevalent among individuals with DCM.[4 10 11] Moreover, a recent study has identified that DCM severely impacts quality of life with recorded 36-Item Short Form Health Survey (SF-36), patient-reported outcome scores among the lowest of all chronic disease.[12] Improving recovery is, therefore, a major unmet clinical need in DCM.[13]

Medical research is primarily designed by healthcare professionals. This bears the risk of not taking into account actual patient needs. The concept of 'research wastage' has emerged to depict healthcare research that does not yield actual or potential clinical benefit. In the 2014 *Lancet* series, Chalmers *et al* estimated that as much as 85% of the US$240 billion expended on health research in 2010 was wasted and an important contributing factor was the misalignment of patient and clinician research objectives.[14 15] As a consequence, several research funding bodies now advocate the involvement of patients in the design and conduct of research. This has demonstrable beneficial impact.[16] Patient and public involvement (PPI) plays a particularly important role in the National Institute for Health Research.[17] In addition to participation and engagement in the research process, the involvement of patients in identifying relevant research topics and their prioritisation is particularly encouraged. Organisations, such as the James Lind Alliance, have successfully brought together patients, professionals and industry in order to set research priorities, for example, for spinal cord injury.[18] However, the research priorities for individuals suffering from DCM have not yet been assessed.

A recent systematic review of DCM research demonstrated a heavy focus on surgical technique.[19 20] However, the research needs of patients with DCM and their priorities remain unknown. Moreover, as part of a core-outcomes initiative REsearch Objectives and COmmon Date Elements in DCM, we have identified that outcome domains are not consistently reported in current clinical research.[19]

In this study, we sought to establish the recovery needs and priorities of individuals suffering from DCM. This will help to determine the outcome assessments that should be included in clinical research and to better direct future research.

## METHODS

Reporting adheres to the Enhancing the QUAlity and Transparency Of health Research (EQUATOR) Network Strengthening the Reporting of Observational Studies in Epidemiologychecklist.[21]

## Survey design

Individuals with DCM and their caregivers were invited to attend the Myelopathy.org PPI day, hosted at the University of Cambridge and captured by Cambridge TV in their documentary.[22] Myelopathy.org is an international, charitable organisation for individuals affected by or working with DCM. As part of the event, qualitative interviews (n=9) were used to establish relevant functional domains that affected quality of life of individuals with DCM. These were found to resemble domains previously reported by Anderson *et al*, who conducted a survey among patients with traumatic spinal cord injury asking them to rank seven domains of spinal cord function in order of priority for recovery.[23] Using this as a template but broadening upper body/trunk strength and balance' to upper body/ trunk function, the following recovery domains were agreed by the participants: elimination of pain, arm and hand function, walking, sexual function, upper body/ trunk function, sensation and bladder/bowel function. For brevity, in this article, they are referred to as arm/ hand, walking, sexual function, pain, sensation, trunk and bladder/bowel.

These questions were embedded into an existing electronic survey initiative, developed using Survey Monkey (California, USA) and following the Checklist for Reporting Results of Internet E-Surveys,[24] investigating patient reporting of DCM. This iteration was piloted by the lead investigators and a selection of individuals with DCM. Study objectives were outlined on the initial page, including details of the host organisation and estimated time required to complete the survey. This acted as the electronic consent, with continuation into the survey as agreement. Respondents were also presented with a description of DCM, including relevant synonyms, and required to confirm they suffered with the condition.

Respondents were asked to rank recovery domains in order of priority and provide details about their DCM. DCM characteristics included age, gender, history of surgery, best daily limb pain score (using a Visual Analogue Scale), duration of symptoms and disease severity as measured using the self-reported, patient-derived, modified Japanese Orthopaedic Association (P-mJOA).[25] The mJOA is among the most commonly used assessments of disease severity[19 20] and is fully-validated.[26] It is a composite score based on upper limb function, lower limb function, upper limb sensation and bladder function. The score is valid for analysis in its entirety or per domain. Originally developed as an investigator-administered tool, it has recently been adapted and validated for use by patients.[25] All questions were mandatory, but respondents were not required to rank every recovery domain, on the basis that some domains may not be a priority for them. The sequence of questions and order of responses was not altered from respondent to respondent.

## Survey administration

The survey was accessed via a landing page on Myelopathy. org, allowing assessment of response rates using Google

Analytics (California, USA). Individuals with DCM were recruited over an 18-month period. The recruitment process has been described in detail previously[27] but in short, the survey was advertised using Google Adwords (California, USA) and through Myelopathy.org and its social media outlets. The survey was voluntary and internet protocol addresses were used to prevent users submitting multiple responses. A missing data analysis was conducted between complete and incomplete survey responses to consider if particular subgroups were more likely to terminate early. Complete responders were defined as having provided answers for all aforementioned variables.

## Analysis

Research priorities are presented using summary statistics, including average ranking and overall proportion of patients per domain. Domains that were not ranked by a respondent were omitted from these scores. For subgroup analysis, variables were dichotomised and thresholds were chosen based on the graphical distribution of responses and sample sizes. Categorical variables were compared using the $\chi^2$ test. For continuous variables, the Shapiro-Wilk test was used to assess for parametric distribution of data sets. The Mann-Whitney U test was then used to compare the means of non-parametric distributions while a two-tailed t-test used to compare the means of parametric distributions. Pearson's correlations were performed to assess between-group differences in characteristics, which could have influenced subgroup analysis. Significance was set at $p<0.05$.

## Patient and public Involvement

Patients were involved in the design, development, recruitment and conduct of this study. At a PPI day hosted at the University of Cambridge, a focus group of DCM patients evaluated and confirmed the recovery domains in DCM. DCM patients were used to pilot the subsequent survey, including optimising its design to reduce the time taken to complete and clarify questions. The online survey for the study was hosted on Myelopathy.org, an international DCM charity run largely by DCM patients. Patients were, therefore, active in disseminating the survey via online DCM support groups, including Myelopathy Support, led by IS, patient and coauthor. Patients who were involved in preparation of the manuscript are among the authors. In addition, all patients who participated in the research are recognised in the acknowledgement statement. DCM patients are involved in plans to disseminate this research to the patient community, including blog articles on Myelopathy.org, posts in online patient support groups and presence at spinal conferences in the UK.

## RESULTS
### Respondents

The survey was uniquely accessed 1463 times, with 659 visitors entering the survey (participation rate of 45%). A

| Table 1 | Summary of respondent demographics |
| --- | --- |
| **Respondent demographics** | |
| Age (Mean±SD) | 53.6 (9.8) |
| Male gender (%) | 140 (29) |
| Undergone surgery (%) | 221 (46) |
| Length of symptoms (%) | |
| 0–1 year | 72 (15) |
| 1–3 years | 140 (29) |
| 3–10 years | 181 (38) |
| 10–25 years | 74 (15) |
| 25+years | 14 (3) |
| P-mJOA (Mean+SD) | |
| Upper limb function | 3.6 (1.0) |
| Walking | 4.4 (1.5) |
| Upper limb sensation | 1.7 (0.7) |
| Bladder function | 2.2 (1.0) |
| Total | 11.9 (3.0) |
| VAS limb pain (Mean±SD) | 3.1 (2.6) |

P-mJOA, patient-derived version of the modified Japanese Orthopaedic Association; VAS, Visual Analogue Scale.

total of 481 responses contained complete data (completion rate 73%). A missing data analysis was conducted comparing incomplete and complete responses. Patients who completed the survey in full were more likely to have undergone surgery ($p$=0.04), otherwise there was no statistical difference within variables of interest (see online supplementary data 1). Only complete responses were analysed in the present study. Of these responses domains were ranked more than 80% of the time: pain (400, 83%), sensation (428, 89%), walking (396, 82%), arm and hand (393, 82%), sexual (388, 81%), bladder and bowel (399, 83%) and trunk function (407, 85%).

On average, respondents were more likely to be female (341, 71%) and suffer with moderate myelopathy (11.9±3.0) for between 3 and 10 years (181, 38%). Around half of patients (221, 46%) had undergone surgery. Overall respondent demographics are summarised in table 1. Considering group differences, patients who had suffered from the disease for longer appeared more likely to have undergone surgery ($p$=0.07) and have worse myelopathy (r=−0.22, $p$<0.005). They were also more likely to suffer greater pain (r=−0.14, $p$<0.01). Average pain scores were 3.1 (±2.4) for patients suffering with the disease for less than a year, rising to 4.5 (±3.0) for patients suffering for at least 10 years. There was no relationship between severity of myelopathy and pain scores (r=−0.04, $p$=0.36). Between-group differences are summarised in online supplementary data 2.

### Ranking of spinal cord dysfunction domains

Overall, pain was the most popular number one ranked recovery domain, chosen by 39.9% of respondents. This

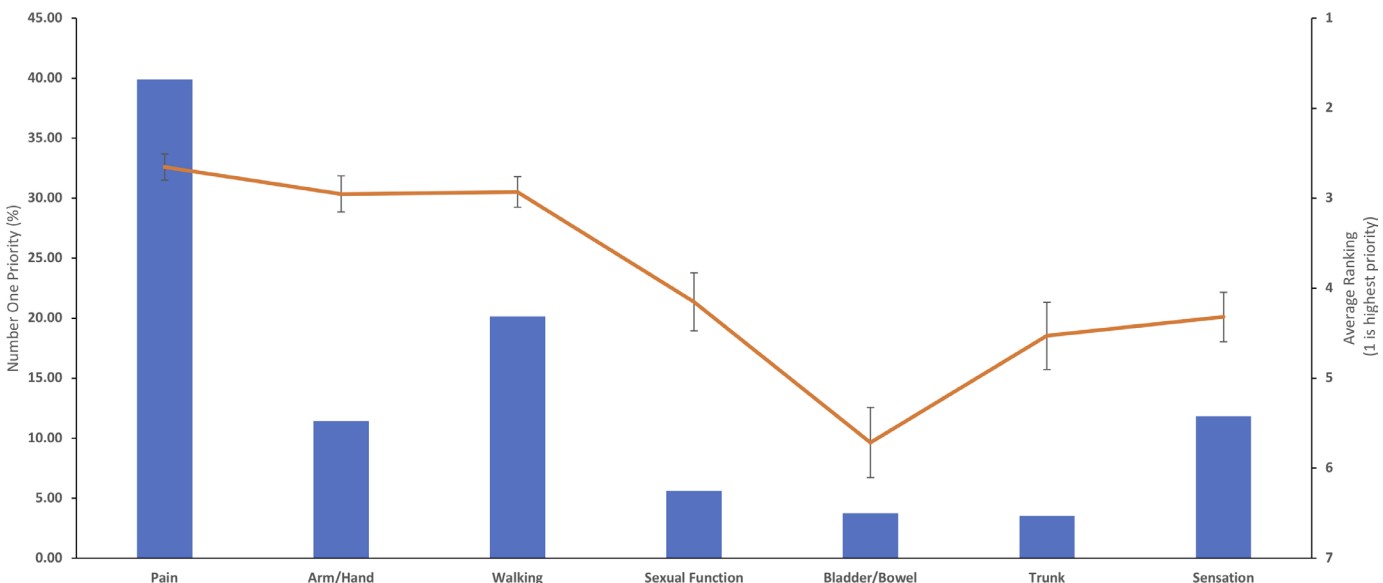

**Figure 1** Overall recovery priorities. The bar chart represents the first choice of patients and the line graph the average ranking for each domain (where the top ranking is 1). Pain was the overall first choice priority of patients, although when priority rankings were averaged, this was closely followed by walking and arm/hand function.

was followed by walking (20.2%), sensation (11.9%) and arm and hand function (11.5%). Sexual function (5.7%), bladder and bowel (3.7%) or trunk function (3.5%) were chosen less frequently. When considering the average ranking of symptoms, while pain remained the priority (2.6±2.0), this was closely followed by walking (2.9±1.7) and arm/hand function (3.0±1.4) (figure 1). Sensation ranked lower (4.3±2.1).

### Impact of baseline characteristics on ranking of spinal cord dysfunction domains

Respondents who had undergone surgery were more likely to prioritise walking ($p<0.005$) and trunk function ($p=0.03$), whereas patients who had not yet undergone surgery were more likely to prioritise upper limb function ($p<0.05$) (figure 2). Patients with poor upper limb or lower limb function were more likely to prioritise arm/hand recovery ($p<0.005$) and walking ($p<0.005$), respectively (figure 2). Overall, pain remained the priority, with the exception of patients with the greatest walking impairment ($p<0.005$), even among patients with lower pain scores (figure 2).

When considering the average rankings pain, arm/hand function and walking remained the top three recovery priorities (figure 3). However, among the subcategories, the order of these priorities differed slightly (see online supplementary data 3). Patients who were male, or who had undergone surgery, or who had greater upper limb, lower limb or bladder functional disability, prioritised recovery of walking, over pain and arm/hand function; patients with greater sensory disability prioritised recovery of arm/hand function over pain and walking.

When overall P-mJOA scores were considered to evaluate mild, moderate and severe patients,[6] no variation was seen in modal or average ranked priorities.

### DISCUSSION

This is the first study to systematically survey functional domains relevant to DCM and to ask patients to rank them in order of importance to their quality of life. The established priorities are likely to reflect symptom prevalence and their impact on day-to-day life.[23] The analysis of 481 completed answers demonstrated that pain, arm/hand function and walking emerge as the most important spinal cord dysfunction domains. Although based on averaged rankings, there were some subtle differences in ordering of these three domains. With the exception of patients with significant gait impairment, elimination of pain was the recovery priority independent of baseline characteristics.

These findings are surprising: functional disability (specifically recovery of arm/hand and walking function) has been and continues to be a focus for researchers, typically in response to surgery,[8] but more recently with a shift towards enhancing postsurgical recovery.[13 27] In contrast, pain is not widely recognised as an important relevant domain. Our recent review of outcome reporting in DCM clinical trials demonstrated that the overwhelming majority of studies (90%) reported outcomes related to function, but only 27% of studies reported outcomes related to pain,[19] despite the fact that pain is a well-recognised feature of DCM,[5] which often improves following surgery.[11] The present findings highlight the fact that systematic research of patient needs is sorely lacking in DCM. A possible explanation for this discrepancy is that surgeons, who play a significant role in the management of DCM and predominate this research field, remain biased towards functional domains because pain is not a recognised indication for surgery in DCM.[6]

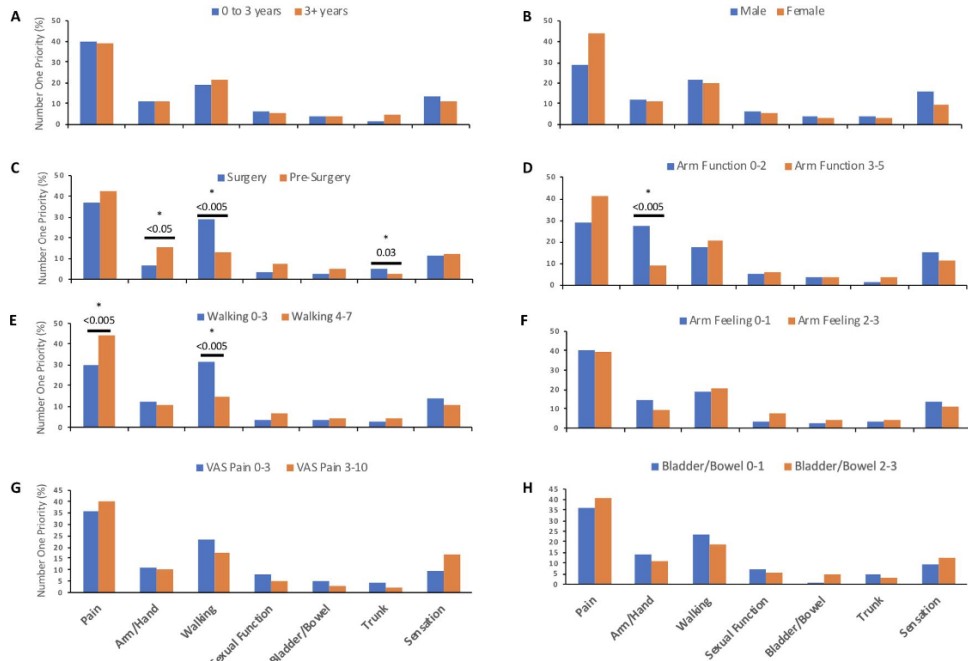

**Figure 2** Impact of baseline characteristics on first choice recovery priority. The bar chart represents the first choice of patients. Significant between-group differences are denoted by the * symbol. For simplicity, groups were dichotomised as follows: (A) duration of symptoms≤3 years, (B) male or female, (C) surgery or pre-surgery, (D) P-mJOA upper limb function≤2, (E) P-mJOA lower limb function≤3, (F) mJOA upper limb sensation≤1, (G) VAS limb pain≤3 and (H) P-mJOA bladder/bowel function≤1 . Those who had undergone surgery were more likely to choose trunk function (*p*=0.03) or walking function (*p*<0.005), whereas those who had not yet undergone surgery were more likely to choose arm/hand function (*p*<0.05). Equally patients with more impairment of walking (*p*<0.005) or arm/hand function (*p*<0.005) were more likely to prioritise these domains. Pain remained the priority even in patients reporting less pain. P-mJOA, patient-derived version of the modified Japanese Orthopaedic Association; VAS, Visual Analogue Scale.

The priorities established in the present study differ from those of individuals suffering from spinal cord injury. Although pain is among the most prevalent symptoms of traumatic spinal cord injury,[28 29] the 'elimination of chronic pain' was considered to be a relatively low priority among those surveyed in Anderson's study[23] and a similar study by Kwon *et al*,[30] that focused on the priorities for SCI recovery after novel treatments (eg, stem cells). Instead, quadriplegics prioritised arm/hand function, while paraplegics sexual and bladder/bowel function. These differences relate to their specific significance for patient independence and quality of life.

In DCM, the symptom burden is less well-described[31 32] and the relationship between symptom burden or significance with respect to quality of life in DCM has not been investigated. However, it would seem likely a similar relationship exists.

### Limitations
Following recommendation by the James Lind Alliance, which was founded to support priority setting in research,[33 34] the present survey was conducted online, as previously described, through a DCM charity, Myelopathy.org.[27] Respondents belonged to a self-selecting group of individuals who were asked to confirm they had been diagnosed with DCM by a medical professional, after being presented with a description of the disease

for verification purposes. It is possible that some respondents did not have DCM. Reassuringly, respondent demographics were comparable to those of leading prospective surgical studies, with the exception of gender, which was not shown to influence patient priorities[8 9] (see online supplementary data 1). This likely reflects the recognised popularity of online health communities among females. There are no such comparable series for non-surgical cohorts, but their inclusion provides a further valuable perspective.

The survey questions were not randomly sorted and therefore each respondent answered identical surveys with spinal cord function domains presented in the same order. The last domain assessed was sensation. In keeping with it being the most prevalent DCM symptom,[32] it featured most frequently in the responses, indicating that the order of domains was unlikely to have influenced the rankings. Moreover, answers to demographic questions, which followed the ranking of priorities in the survey, were required to define a complete response in order to be included in the present analysis. Priorities, therefore, were not influenced by incomplete answers.

Following the qualitative development work and the previous experience of Anderson *et al*, the pain domain was kept non-specific, asking patients to rank 'elimination of pain' as a recovery priority (see online supplementary

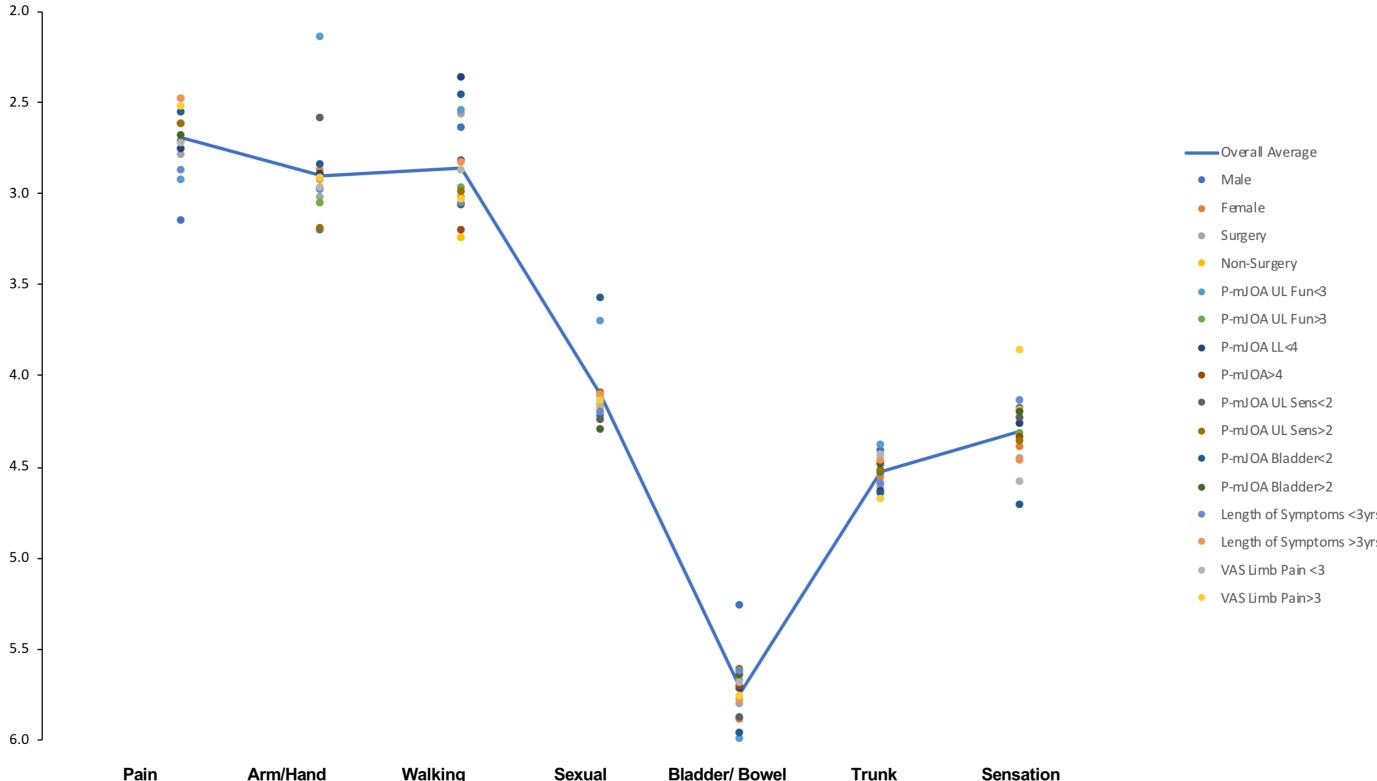

**Figure 3** Impact of baseline characteristics on recovery priority average rankings. The scatter plot represents the mean ranking for each subgroup investigated. The blue line represents the overall average. Despite some discrepancies between subgroups, pain, arm/hand and walking function were consistently the top three priorities for patients. Bladder/bowel function was not a recovery priority. P-mJOA, patient-derived version of the modified Japanese Orthopaedic Association; VAS, Visual Analogue Scale.

data 4). In contrast, however, the pain assessment focused on limb pain, which is classically felt to represent DCM-related pain.[5] While this does not limit the implications of our findings as whole, their interpretation will require a better characterisation of pain in DCM in order to focus research appropriately as other pain foci are reported.[35]

## CONCLUSION

The priorities reported in the present study identify functional domains that are relevant to the quality of life of DCM patients. They provide an important framework for future research and will serve as a valuable reference for the development of a core outcome set relevant to studies in DCM.

**Author affiliations**
[1]Division of Neurosurgery, Department of Clinical Neurosciences, University of Cambridge, Cambridge, UK
[2]Myelopathy.org, Cambridge, UK
[3]Division of Neurosurgery, University of Maryland, Baltimore, Maryland, USA
[4]Department of Orthopaedics, University of British Columbia, Vancouver, UK
[5]Department of Neurosurgery, Medical College of Wisconsin, Milwaukee, Wisconsin, USA
[6]Department of Neurosurgery, Thomas Jefferson University Hospital, Philadelphia, Pennsylvania, USA
[7]Department of Neurosurgery, Toronto Western Hospital, Toronto, UK
[8]Department of Neurosurgery, Houston Methodist Hospital, Houston, Texas, USA
[9]Anne McLaren Laboratory for Regenerative Medicine, Department of Clinical Neurosciences, University of Cambridge, Cambridge, UK

**Acknowledgements** The authors are grateful to all DCM patients who participated in this research.

**Contributors** BD, OM, IS, BA, BK, SK, JSH, JRW, RG, MGF and MK were involved in the interpretation, drafting and final approval of the manuscript. Additionally, authors BD, IS and MK were involved in the conception, design and acquisition of data for this study, whilst authors BD and OM conducted the data analysis.

**Funding** This report is independent research arising from a Clinician Scientist Award, CS-2015- 15-023, supported by the National Institute for Health Research. Research in the senior author's laboratory is supported by a core support grant from the Wellcome Trust and MRC to the Wellcome Trust-Medical Research Council Cambridge Stem Cell Institute. MK is supported by an NIHR Clinician Scientist Award. MGF is supported by the Halbert Chair in Neural Repair and Regeneration. Partial support for this work was obtained from the AOSpine Knowledge Forum in Spinal Cord Injury. The authors also acknowledge the AOSpine for their support of travel and meetings costs. The NIHR HTC, from the Brain Injury Medical Technology Cooperative, provided the funding for Google Adwords survey advertising.

**Disclaimer** The views expressed in this publication are those of the authors and not necessarily those of the NHS, the National Institute for Health Research or the Department of Health and Social Care.

**Competing interests** JSH reports being a Medical Advisor for Depuy Synthes and Ethicon, being an Educational Speaker at Globus Medical and research funding from AO Spine. MGF reports consulting for Fortuna Fix. MK declares a grant from the National Institute for Health Research, travel support from AO Spine and is founder of Myelopathy.org, the first charity for patients with cervical myelopathy. The remaining authors have nothing to declare.

**Patient consent for publication** Not required.

**Ethics approval** Ethical approval was granted by the University of Cambridge.

**Provenance and peer review** Not commissioned; externally peer reviewed.

**Data availability statement** All data relevant to the study are included in the article or uploaded as online supplementary information.

**ORCID iD**

Oliver Mowforth http://orcid.org/0000-0001-6788-745X

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
