## [Reviewer comments · BMJ Open]

ARTICLE DETAILS

TITLE (PROVISIONAL)	Recovery priorities in degenerative cervical myelopathy: a cross-sectional survey of an international, online community of patients
AUTHORS	Davies, Benjamin; Mowforth, Oliver; Sadler, Iwan; Aarabi, Bizhan; Kwon, Brian; Kurpad, Shekar; Harrop, James; Wilson, Jefferson R.; Grossman, Robert; Fehlings, Michael G.; Kotter, Mark

VERSION 1 – REVIEW

REVIEWER	Andrei F Joaquim University of Campinas (UNICAMP), Campinas, Brazil
REVIEW RETURNED	09-Jun-2019

GENERAL COMMENTS	the manuscript is well written and very interesting, once its evaluate patients recovery priorities instead of researcher outcomes. I congratulate the authors for this initiative
---

REVIEWER	Atsushi Kimura Jichi Medical University, Japan
REVIEW RETURNED	13-Jun-2019

GENERAL COMMENTS	This study investigates patient priorities of symptoms in DCM using an internet-based survey. The authors demonstrate that pain was the most popular recovery priority of respondents, followed by walking, sensation and arm and hand function. There are several concerns that would have to be addressed. 1. The authors state that mJOA score is fully validated outcome (page 7). However, as far as I know, the mJOA is an investigator-administered tool and is not validated as a patient-reported outcome. More detailed explanations are required for this point.2. Figure 1 shows the most important results of this study. However, in my opinion, this figure seems to be hard to comprehend. I find limited scientific validity in these results, especially in the ranking, because the sequence of questions and order of responses was not altered from respondent to respondent (page 7). In addition, I think it is uncommon and confusing to show this type of data as a line plot.3. As an important limitation of this study, sampling bias resulting from the nature of internet-based survey need to be discussed in more detail.
--

VERSION 1 – AUTHOR RESPONSE

Reviewer: 1

Reviewer Name

Andrei F Joaquim

Institution and Country

University of Campinas (UNICAMP), Campinas, Brazil

Please state any competing interests or state 'None declared':

None declared

Please leave your comments for the authors below
the manuscript is well written and very interesting, once its evaluate patients recovery priorities instead of researcher outcomes.

I congratulate the authors for this initiative

A: We thank Dr Joaquim for reviewing our manuscript and for the complements on the study. We agree that studies focusing on patients' perspective of important outcomes in DCM are greatly needed in a field that focuses on researcher-selected outcomes.

Reviewer: 2

Reviewer Name

Atsushi Kimura

Institution and Country

Jichi Medical University, Japan

Please state any competing interests or state 'None declared':

None declared

Please leave your comments for the authors below
This study investigates patient priorities of symptoms in DCM using an internet-based survey. The authors demonstrate that pain was the most popular recovery priority of respondents, followed by walking, sensation and arm and hand function. There are several concerns that would have to be addressed.

A: We thank Dr Kimura for reviewing our manuscript.

1. The authors state that mJOA score is fully validated outcome (page 7). However, as far as I know, the mJOA is an investigator-administered tool and is not validated as a patient-reported outcome. More detailed explanations are required for this point.

A: The mJOA has recently been adapted from an investigator-administered tool, to a patient reported outcome measure by Rhee et al., (Clin Spine Surg. 2018 Mar) [25]. They named the tool the p-mJOA. It was validated using a prospective cohort study. The study found that p-mJOA and mJOA had identical mean scores in assessing myelopathy, with moderate to strong agreement between the two scoring systems across a range of measures of reliability. The p-mJOA was also found to have lower patient burden than the mJOA and was preferred by patients. Whilst this was published in 2018, the tool was made available to Dr Kotter, senior author, by personal communication, for use in this study. We have added a summary of the above in the methodology to make this clear for the reader. We have also altered the acronym from mJOA to P-mJOA.

2. Figure 1 shows the most important results of this study. However, in my opinion, this figure seems to be hard to comprehend. I find limited scientific validity in these results, especially in the ranking, because the sequence of questions and order of responses was not altered from respondent to respondent (page 7). In addition, I think it is uncommon and confusing to show this type of data as a line plot.

A: We thank Dr Kimura for highlighting this point. Whilst we do agree that not changing the order of questions from respondent to respondent has the potential of introducing question order bias, we discuss why we do not believe this to be the case in this study within the discussion section of the manuscript. In short, the last domain presented in the questionnaire was always sensation, which was the domain which featured most, not least, commonly in the responses. This suggests that domains towards the end of the survey were not less likely to be ranked than domains at the beginning of the survey. Moreover, questions on demographics were after the questions on the recovery domains, meaning that all recovery domain questions were towards the beginning of the survey and only responses which were fully complete were included in the final analysis. A missing data analysis suggested exclusion of non-complete responses did not introduce additional bias. This is addressed in paragraph 2 of the limitations section.

We acknowledge that Figure 1 is a complex figure that summarises important data. The aim was to show that whilst pain was the number one priority (modal first priority - bar chart), walking and hand function were also patient priorities (mean ranking - line graph). After careful reflection we remain of the opinion that this is the most effective way to present this data (although would of course be happy to consider any alternative suggestions). It should be noted that Figure 1 was well-received when the work was presented to audiences at a national and an international neurosurgical conference.

3. As an important limitation of this study, sampling bias resulting from the nature of internet-based survey need to be discussed in more detail.

A: We thank Dr Kimura for this important suggestion – it is certainly an important area to consider when interpreting survey data, especially in the context of an open-access internet survey. We have addressed this in paragraph 1 of the limitation section, where we note the following features:

- 1) The use of open-access internet surveys is established for priority setting process, as outlined by the National Institute for Health Research James Lind Alliance
- 2) Recruitment was conducted via Myelopathy.org, an international charity for DCM
- 3) For verification purposes, respondents were asked to confirm that they had been given a diagnosis of DCM by a medical professional, having been provided with a definition of DCM, before accessing the survey.
- 4) For external validation, the demographics of respondents were compared to the AO Spine clinical datasets of patients undergoing surgery and were comparable.

In short, we accept that internet surveys have their limitations, but we have taken steps to minimise these, and evaluate their influence in our findings, whilst being transparent about their possibility.

However whilst the use of an internet survey has limitations, it should be recognised that it is the reach of the internet that has enabled this unique perspective: this is the first study of its kind in DCM, the largest ever on patient perspective in myelopathy and includes both surgical and non-surgical patients. This latter point is significant, as non-surgical cohorts are under-represented in DCM research.

We thank Dr Kimura for reviewing our manuscript and hope we have provided adequate reassurance. We have made a number of changes to the manuscript to ensure this information is clearer for readers.

VERSION 2 – REVIEW

REVIEWER	Atsushi Kimura Department of Orthopaedics, Jichi Medical University, Japan.
REVIEW RETURNED	21-Jul-2019
GENERAL COMMENTS	This paper was adequately revised.